psychology

demand characteristics, phenomenological control, rubber hand illusion, embodiment, imaginative suggestion, hypothesis awareness

**Author for correspondence:**
P. Lush
e-mail: p.lush@sussex.ac.uk

# Hypothesis awareness confounds asynchronous control conditions in indirect measures of the rubber hand illusion

## P. Lush[1,2], A. K. Seth[1,2] and Z. Dienes[1,3]

[1]Sackler Centre for Consciousness Science, University of Sussex, Falmer, BN1 9RH, UK
[2]Department of Informatics, University of Sussex, Chichester Building, Falmer, BN1 9RH, UK
[3]Department of Psychology, University of Sussex, Pevensey Building, Falmer, BN1 9RH, UK

PL, 0000-0002-0402-1699; AKS, 0000-0002-1421-6051; ZD, 0000-0001-7454-3161

Reports of changes in experiences of body location and ownership following synchronous tactile and visual stimulation of fake and real hands (rubber hand (RH) effects) are widely attributed to multisensory integration mechanisms. However, existing control methods for subjective report measures (asynchronous stroking and control statements) are confounded by participant hypothesis awareness; the report may reflect response to demand characteristics. Subjective report is often accompanied by indirect (also called 'objective' or 'implicit') measures. Here, we report tests of expectancies for synchronous 'illusion' and asynchronous 'control' conditions across two pre-registered studies (n = 140 and n = 45) for two indirect measures: proprioceptive drift (a change in perceived hand location) and skin conductance response (a measure of physiological arousal). Expectancies for synchronous condition measures were greater than for asynchronous conditions in both studies. Differences between synchronous and asynchronous control condition measures are therefore confounded by hypothesis awareness. This means indirect measures of RH effects may reflect compliance, bias and phenomenological control in response to demand characteristics, just as for subjective measures. Valid control measures are required to support claims of a role of multisensory integration for both direct and indirect measures of RH effects.

# 1. Hypothesis awareness confounds asynchronous control conditions in indirect measures of the rubber hand illusion

Demand characteristics are cues which inform beliefs regarding experimental aims to participants and therefore influence experimental results [1]. Such cues are not limited to experimental instruction but can arise from any aspect of the experimental situation, including the pre-existing beliefs of participants. Demand characteristics can lead to hypothesis awareness when participant expectancies match experimenters' predictions. Hypothesis awareness effects include faking, and imagination which can generate false positives, but also reactance, which can lead to false negatives (see [2] for a recent conceptual model of demand characteristics). A particular concern for rubber hand (RH) illusion measures is the possibility that they reflect implicit imaginative suggestion effects [3,4] or 'phenomenological control' (see [5,6]).

RH effects involve experiences of ownership and feelings of mislocated touch when a fake hand (which is in view) and the participant's own hand (which is hidden from view) are brushed in synchrony. Typically, these experiences are thought to be driven by multisensory integration mechanisms ([7]; see [8] for a review). Proponents of this view generally refer to these effects as the RH illusion. However, there are dissenting views. For example, Dieguez [9] argued that the RH illusion should not be considered an illusion in the same sense as classic visual or optical illusions, as it is likely to arise from participant expectancies (we agree, and henceforth employ inverted commas when referring to RH effects as illusions), and Alsmith [10] argues that RH experience may reflect imaginative experiences. Consistent with these accounts, we have previously shown that demand characteristics are not controlled in subjective reports of RH effects [11], and that reports of RH experiences are substantially predicted by phenomenological control. For example, trait response to an imaginative suggestion on a 6-point scale predicts subjective reports of ownership experience on a 7-point scale by 0.8 units per scale point ([12]; see also [6,13]).

Here, we present an investigation of the validity of RH effect control methods for two indirect or 'implicit' measures: proprioceptive drift (a reported shift in the perceived location of the participant's own hand first reported by [7]) and skin conductance response (SCR) changes in electrical properties of the skin which can accompany changes in arousal; (e.g. [14,15]).

In RH research, subjective reporting is commonly measured by Likert scale responses to three statements describing referred touch and ownership experience, with agreement recorded on a 7-point scale from −3 to +3 (negative values indicate disagreement). These reports are generally taken after the induction procedure has ceased. A set of statements describing other experiences are often included as control statements. See table 1 for statements and scale labels. Proprioceptive drift is typically measured by reports of the perceived position of the participant's (unseen) hand before and after: the stroking procedure. See Riemer *et al.* [8] for a review of RH effect procedures.

In addition to the experimental condition in which the brush strokes on the real and fake hands are performed in synchrony, an asynchronous condition in which there is a delay between visual and tactile stimulation is commonly employed as a control. For subjective RH reports, we have shown that control statements and the asynchronous control condition are invalid controls because the differences between both 'illusion' and control statement response and synchronous and asynchronous condition response are confounded by expectancies [11]. By contrast with direct subjective reports of experience, proprioceptive drift and skin conductance are indirect measures of RH effects. Although these indirect measures are often described as 'objective' or 'implicit' measures, it is plausible that, like subjective report, they may be susceptible to demand characteristics. It is also common to employ an asynchronous control condition when interpreting indirect measures (see [8]). An underlying assumption of an experimental control is that it holds everything constant except the independent variable relating to the mechanism of interest. In RH effect studies, the mechanism of interest is typically multisensory integration, and the asynchronous control condition is intended to keep all factors constant except for the timing of multisensory stimuli. However, participant expectancies may differ for these conditions. If so, this crucial assumption would be violated and the control procedure would not therefore be valid (because any difference between conditions may be attributable to a difference in expectancies instead of or in addition to differences in the timing of multisensory stimuli). It is therefore crucial to establish whether participants have differing expectancies for indirect measures in synchronous and asynchronous conditions.

There is ongoing debate regarding the degree to which proprioceptive drift is related to subjective reports of RH effects (see [8]). There is evidence that proprioceptive drift can dissociate from

**Table 1.** Statements, questions and response labels are used to generate subjective report scores. All statements were taken from Botvinick & Cohen [7].

| illusion and control statements | scale labels |
| --- | --- |
| **illusion** | |
| S1. It seemed as if I were feeling the touch of the paintbrush in the location where I saw the rubber hand touched | +3. I am certain I will feel some effect |
| S2. It seemed as though the touch I felt was caused by the paintbrush touching the rubber hand | +2. I am fairly certain I will feel some effect |
| S3. I felt as if the rubber hand were my hand | +1 I think I will feel some effect |
| **control** | 0. I have no idea either way |
| C1. It felt as if my (real) hand were drifting toward the rubber hand | −1. I think I won't feel any effect |
| C2. It seemed as if I might have more than one left hand or arm | −2. I am fairly certain I won't feel any effect |
| C3. It seemed as if the touch I was feeling came from somewhere between my own hand and the rubber hand | −3. I am certain I won't feel any effect |
| C4. It felt as if my (real) hand were turning 'rubbery' | |
| C5. It appeared (visually) as if the rubber hand were drifting towards the left (towards my hand) | |
| C6. The rubber hand began to resemble my own (real) hand, in terms of shape, skin tone, freckles or some other visual feature. | |

subjective reports (e.g. [16]), and that it can occur in the asynchronous condition (e.g. [16,17]). However, it is a commonly referenced measure of RH effects (Google Scholar returns 1420 results for 'proprioceptive drift', 9 April 2021) and is sometimes presented without any accompanying subjective report (e.g. [18]). Proprioceptive drift is generally considered an indirect measure because it requires no direct introspection regarding ownership or felt touch. Rather, it requires introspection regarding the perceived location of an unseen limb. In this way, it is similar to indirect measures of response to imaginative suggestion which involve reports of perceived limb location following imaginative suggestion (e.g. that one's hands are drawn together by a magnetic force; [19,20]). Like subjective reports, proprioceptive drift is related to trait response to imaginative suggestion (e.g. [6,21]). These relationships can be substantial; in a sample of 353 people, trait response to imaginative suggestions of a range of experiences (e.g. visual and auditory hallucinations) predicts a drift of 0.6 cm for each point on a 6-point scale of response to suggestion [6]. In a previous investigation of demand characteristics in RH effects, Tamè et al. [17] reported a weaker magnitude of drift (and subjective reports) when participants were asked to report where their hand 'really is' rather than where it 'feels like' it is, and argued that proprioceptive drift may be attributable to demand characteristics. There are reports of proprioceptive drift in unimodal cases, for example, when a fake hand is merely viewed [16] or when brushes are replaced with lasers [22]. Such cases have been interpreted as evidence for multi-modal integration effects either because imagined tactile experience is interpreted as a sensory modality [22] or because drift may reflect the integration of proprioceptive and visual information [16]. However, these interpretations also require ruling out the influence of demand characteristics. In an exploration of order effects, participants who underwent the asynchronous control condition before the synchronous condition reported a greater difference between synchronous and asynchronous conditions [23]. This result may indicate hypothesis awareness arising from task order. Together, these observations are consistent with the proposal that proprioceptive drift may reflect effects driven by expectancies arising from demand characteristics.

SCR is typically measured in RH studies by presenting a threat to the fake hand during or following induction, again controlled by comparison with an asynchronous stroking condition (e.g. [14]). Although some RH researchers assert that SCR cannot be voluntarily controlled, [14,24], there is a substantial body of research demonstrating voluntary control of SCR (e.g. [25]; for a review see [26])

including in response to imaginative suggestion (across more than half a century, e.g. [27,28]). Notably, changes in SCR occur when participants are prompted to imagine that virtual arms presented on a screen are their own [29]. As with proprioceptive drift, these observations suggest that SCR may be sensitive to demand characteristics.

Here, in two pre-registered studies, we replicate previous results showing expectancies for subjective report of RH effects differ for synchronous and asynchronous conditions [11] and extend investigation of expectancies for synchronous and asynchronous conditions to the indirect RH effect measures of proprioceptive drift and SCR. As Orne [30] noted, expectancy studies (or 'pre-experimental enquiries') can never provide evidence that a given effect is attributable to demand characteristics. Rather they test the adequacy of an experimental procedure for controlling the effects of demand characteristics. If an experimental procedure is inadequate for controlling, for example, hypothesis awareness, it follows that hypothesis awareness effects cannot be ruled out for studies which employ that procedure. If participant expectancies for synchronous and asynchronous conditions differ in the direction reported in RH experiments, any difference in these measures may be attributable to demand characteristics and consequently, existing reports of proprioceptive drift and SCR may reflect phenomenological control (or other hypothesis awareness effects including imagination and faking; see [2]) rather than, or in addition to, multisensory processes. This would have major implications for interpretation of existing reports of these effects because we would be unable to disentangle hypothesis awareness effects from other effects in any given case.

# 2. Study 1: proprioceptive drift expectancies

## 2.1. Method

### 2.1.1. Participants

Data from 172 participants were recorded. Thirty-one participants were undergraduate psychology students recruited from the University of Sussex recruitment database and 141 were current UK psychology undergraduates fluent in English recruited on Prolific (https://www.prolific.co/). Sussex participants were compensated with course credit. Prolific participants were compensated with a payment of £1.50. In accordance with pre-registered exclusion criteria (preregistration document available at https://osf.io/9c8mq), 32 participants were excluded, 17 for spending less than ten seconds reading the information page and 15 for reporting previous participation in a procedure similar to that shown in the video. Because Bayes factors for each pre-registered analysis were greater than the pre-registered stopping rule threshold (greater than 6 or less than 1/6), data collection ceased before data from 200 participants had been collected. Data from 140 participants (106 female, 34 male) with a mean age of 24.1 (s.d. = 6.8) were therefore included. For a pre-registered subgroup analysis, 67 participants who reported having heard of the procedure before were further excluded, leaving 73 participants (56 female, 17 male) with a mean age of 24.6 (s.d. = 7.6). All participants provided informed consent and ethical approval was granted by the University of Sussex ethics committee.

### 2.1.2. Procedure (adapted from [11])

All study materials are available at https://osf.io/ct7qe/. Participants completed the study online using their own computers. After providing consent and demographic information, participants were asked to read the following short passage describing the RH procedure:

> 'In this procedure, a participant's own hand is hidden from their view and a fake hand is placed in front of them. An experimenter then uses brushes to stroke the participants hidden real hand and the visible fake hand. The location of the brush strokes on the real and fake hands is matched, so that a downward brush stroke on the participants index finger will be accompanied by a downward brush stroke on the fake hand. Participants can therefore see a paintbrush brushing down the finger on a fake hand while they feel a paintbrush brushing down the finger on their real hand (which they cannot see). There are two conditions in the experimental procedure: Synchronous condition: The brush strokes on the real hand and on the fake hand occur at the same time (they are synchronous). Asynchronous condition: The brush strokes on the participants real hand and on the fake hand occur at different times (they are asynchronous)'

Participants were then shown a 62 s video in which the synchronous and asynchronous procedures were demonstrated. This was followed by a brief passage of text describing the procedures shown in the video:

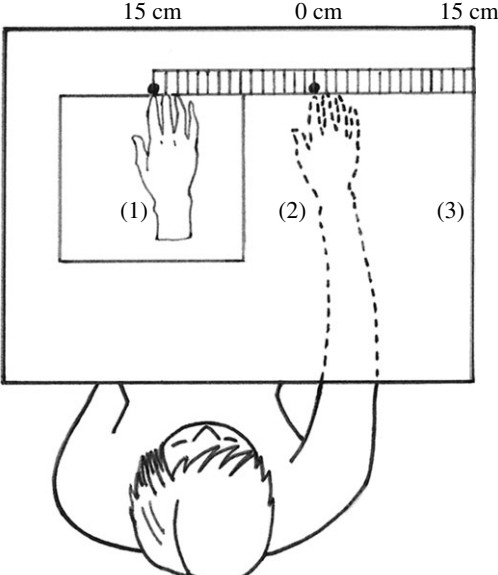

**Figure 1.** Illustration provided to participants. The figure was accompanied by the following legend: the RH (1) is positioned to the left of the participant's hand (2). The participant cannot see their own right hand (2), which is inside the box (dashed line). The participant can see the RH (1) through a window in the box. The index finger of the RH (1) is positioned 15 cm to the left of the participant's right index finger (2). The edge of the box (3) is 15 cm to the right of the participant's right index finger.

> 'In the video, an experimenter performed brush strokes on the participant's hand and a fake hand in matching locations (the index fingers) on each hand. The participant could see the fake hand but could not see their real hand. There were two conditions. In the synchronous condition, the brush strokes on the real and fake hands occurred at the same time. In the asynchronous condition there was a delay between the brush strokes on the fake hand and the real hand.'

Participants were then asked to briefly answer the following question which was intended to provoke consideration of the previously presented information: 'What do you think this procedure is supposed to cause (what is the participant expected to experience)?' and asked whether or not they had heard of the procedure before and whether or not they had previously participated in an experiment in which the procedure was used.

The following was added to the procedure reported in Lush [11] to measure proprioceptive drift expectancies. A figure illustrating a birds-eye view of the experimental set-up was displayed with a legend describing a set-up for proprioceptive drift measurement (figure 1; based on the procedure in [31] and [6]).

Participants then read the following description of a proprioceptive drift measurement procedure in which estimates of the position of the participant's hand are recorded before brush stroking and following brush stroking in the synchronous and asynchronous conditions:

> 'At the beginning of the experiment, the participant cannot see their own right hand or the fake hand. The experimenter then places a long ruler horizontally across the box and asks the participant to use their left hand to point to the position on the ruler which is directly above their right index finger (which is hidden inside the box).
>
> The experimenter then removes the cover from the box so the participant can see the fake hand and asks them to focus on the fake hand. The experimenter then performs the procedure shown in the video (strokes the participant's hand and the fake hand either synchronously or asynchronously).
>
> After 1 min of the brush stroking procedure, the experimenter covers the hole on the box so that the participant can no longer see the fake hand and asks them to keep their right hand still inside the box. The experimenter once again places the ruler across the box and asks the participant to use their left hand to point to the position on the ruler which is directly above their right index finger (which is hidden in the box).
>
> Note that the participant's own hand is not visible at any point (it remains concealed inside the box).'

This was followed by a free report question which was again intended to provoke consideration of the presented material: 'Why do you think the experimenter asks the participant to estimate the location of their right index finger before and after the procedure?'.

To record expectancies for perceived hand position at baseline (before stroking) and following synchronous and asynchronous stroking conditions, participants used a horizontal slider labelled from −15 cm to +15 cm to report where they think they would judge their index finger to be if they were a participant in the described experiment. They were informed that 15 cm represents the position of the fake hand, 0 cm the actual position of the participant's index finger inside the box and −15 cm the edge of the box (away from the RH). Participants were asked to use the slider to report 'where you think you would judge your index finger to be if you were a participant in this procedure'. Judgements for baseline (before stroking) and following synchronous and asynchronous brush stroking were recorded on subsequent screens.

Finally, participants reported expectancies for synchronous and asynchronous conditions for each of the nine statements used in the original RH study, in a fixed order (as in [11]). Table 1 shows the illusion and control statements taken from Botvinick & Cohen [7] and the scale labels used to measure expectancies for each statement. The seven-item scale is taken from Lush *et al.* [20] and is based on the seven-point scale which measures agreement and disagreement with RH effect statements introduced by Botvinick & Cohen [7].

### 2.1.3. Measures

A proprioceptive drift expectancy measure was calculated for synchronous and asynchronous conditions from the difference between baseline judgement and synchronous or asynchronous condition judgements.

Response to the three 'illusion' statement expectancies (S1–S3) was used to calculate a mean 'illusion' expectancy score for both synchronous and asynchronous conditions. Response to the six 'control' statements (C1–C6) was used to calculate a mean 'control' expectancy score for the synchronous condition.

### 2.1.4. Pre-registered analyses

Study 1 analyses are pre-registered at https://osf.io/98xyh. Differences between synchronous condition and asynchronous condition expectancies in the full sample and in a subsample of participants who reported that they had not heard of the procedure were analysed with *t*-tests. For proprioceptive drift expectancies, Bayes factors were calculated with H1 modelled by a half normal based on the 1 cm difference in proprioceptive drift between synchronous and asynchronous induction reported in Botan *et al.* [31] and as in [6]. For the subjective report, Bayes factors were calculated as in [11], using a half normal based on the 1 scale point difference in expectancy between synchronous and asynchronous induction reported in Lush *et al.* [6]; 95% CIs (interpreted as Bayesian credibility intervals with a uniform prior) were used to estimate measures. Bayes factors were calculated with the calculator at: https://harry-tattan-birch.shinyapps.io/bayes-factor-calculator/. Robustness regions (RR) were determined as the set of scale factors that led to the same qualitative conclusion (either $B > 3$, or $B < 1/3$; or $1/3 < B < 3$; [9]).

Bayes factors were used to assess the strength of evidence for H1 versus H0 after the first 20 participants (following exclusion) and data collection ceased when evidence was sensitive in either direction (Bayes factor of greater than 6 or less than 1/6).

## 2.2. Results

Figure 2 shows expectancy ratings for synchronous and proprioceptive drift in the whole sample. Expected proprioceptive drift was higher for the synchronous condition ($M = 4.5$ cm, s.d. = 6.0) than for the asynchronous condition ($M = 1.7$ cm, s.d. = 6.1), $t_{139} = 5.62$, $p < 0.001$, 95% CI [1.9, 3.9], $d = 0.48$ 95% CI [0.30, 0.65], $B_{H(0,1)} = 2.39 \times 10^5$ RR$_{B>3}$ [0.1, >30].

Figure 3a shows mean illusion subjective report expectancy ratings and figure 3b shows mean expectancy ratings for individual statements in the whole sample. Replicating previous results [11], synchronous condition illusion (S1–S3) expectancies ($M = 1.7$, s.d. = 1.0) were higher than for the asynchronous condition ($M = 0.2$, s.d. = 1.2), $t_{139} = 11.91$, $p < 0.001$, 95% CI [1.3, 1.8], $d = 1.01$ 95% CI [0.81, 1.21], $B_{H(0,1)} = 9.78 \times 10^{28}$ RR$_{B>3}$ [0.02, >6]. Synchronous condition illusion expectancies were also higher than synchronous condition control (C1–C6) statement expectancies, ($M = 0.1$, s.d. = 1.1), $t_{139} = 15.6$, $p < 0.001$, 95% CI [1.4, 1.8], $d = 1.32$ 95% CI [1.09, 1.55], $B_{H(0,1)}$ $9.37 \times 10^{29}$ RR$_{B>3}$ [<0.01, >6] figure 3.

In participants who reported not having heard of the presented RH procedure, expected proprioceptive drift was higher in the synchronous ($M = 3.7$ cm, s.d. = 7.2) than asynchronous condition ($M = 1.7$ cm, s.d. = 7.0). $t_{72} = 2.80$, $p = 0.007$ 95% CI 0[0.6, 3.5], $d = 0.33$ 95% CI [0.09, 0.56], $B_{H(0,1)}$. 14.91 RR$_{B>3}$ [0.33, 24.0].

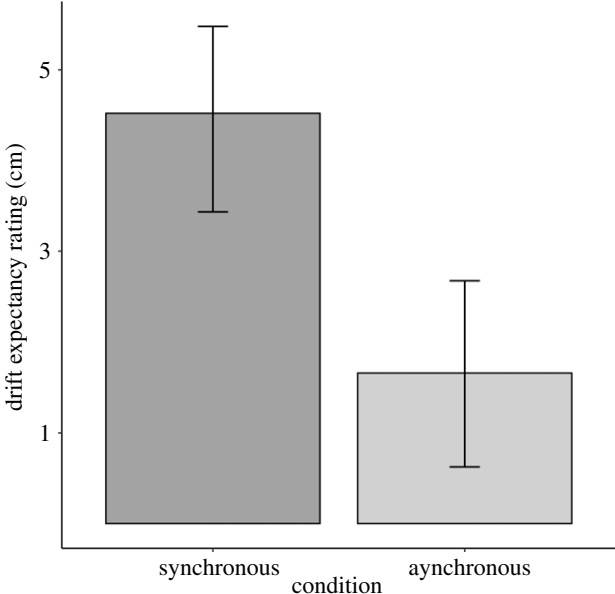

**Figure 2.** Mean proprioceptive drift expectancies for synchronous and asynchronous conditions. Error bars show 95% CIs.

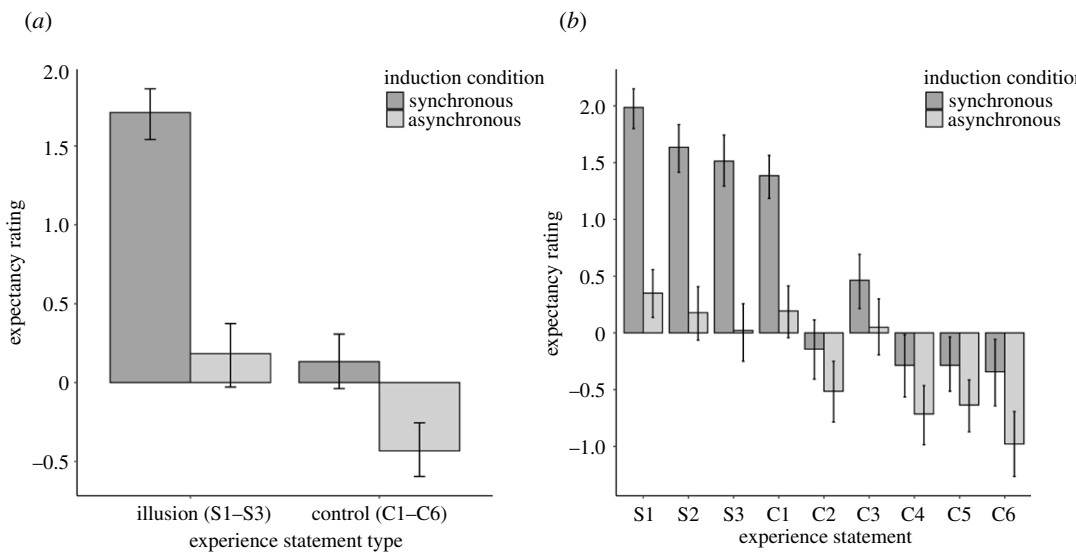

**Figure 3.** (*a*) Subjective report expectancies for the mean illusion (S1–S3) and control (C1–C6) statements for synchronous and asynchronous conditions. Error bars show 95% CIs. (*b*) Subjective report expectancies for individual statements for synchronous and asynchronous conditions. Error bars show 95% CIs.

Synchronous illusion expectancy rating ($M = 1.7$, s.d. $= 0.90$) was greater than asynchronous illusion expectancy rating ($M = 0.2$, s.d. $= 1.2$) in participants who reported no experience of the illusion, $t_{72} = 8.49$, $p < 0.001$, 95% CI [1.2, 1.9], $d = 0.99$ [0.71, 1.27], $B_{H(0,1)}$ $1.43 \times 10^{14}$ $RR_{B>3}$ [0.03, >6]. Synchronous illusion expectancy rating was also greater than synchronous control expectancy rating ($M = 02$, s.d. $= 1.0$) in this sub-group., $t_{72} = 12.06$, $p < 0.001$, 95% CI [1.2, 1.7], $d = 1.41$ [1.08, 1.73], $B_{H(0,1)} = 5.97 \times 10^{29}$ $RR_{B>3}$ [0.02, >6].

## 3. Study 2: skin conductance response expectancies

A second study was conducted to test expectancies for SCR. We predicted that participants would estimate measures of physiological signals to be greater in the synchronous than asynchronous condition.

## 3.1. Method

### 3.1.1. Participants

Fifty participants fluent in English and resident in the UK were recruited using Prolific (https://www.prolific.co/). Five participants were excluded according to pre-registered criteria: one for spending less than an average of 4 s on the nine agreement statements reports, two for spending less than 10 s on the information page and two for reporting previous participation in a RH study. Data from 45 participants (12 male, 33 female) with a mean age of 32.0 (s.d. = 13.0) were therefore analysed. For pre-registered sub-group analyses, 33 participants who reported not having heard of the procedure were included (24 female, 9 male) with a mean age of 33.8 (s.d. = 13.6). For exploratory sub-group analyses, data from 31 participants who reported not having studied psychology at degree level and not having heard of the procedure were analysed (22 female, 9 male) with mean age 34.3 (s.d. = 13.8).

### 3.1.2. Procedure

The procedure replicated Study 1, with three additional questions. First, participants were asked whether or not they are currently studying for or have completed a degree in psychology. Second, they were asked to provide a free report regarding what they thought would be measured by electrodes attached to their hand during the brush stroking procedure (see following section). Finally, they were asked to report whether they thought the experimenter would expect higher measurements from the electrodes for synchronous than asynchronous conditions. Material regarding skin conductance was presented after proprioceptive drift and subjective report expectancies had been recorded.

### 3.1.3. Materials

The study materials for measuring proprioceptive drift and subjective report expectancies were as in Study 1. For skin conductance expectancies, participants were presented with the following text:

> 'Before the procedure, electrodes are attached to your left hand and connected to a computer. During the procedure a second experimenter suddenly stabs the rubber hand with a knife'.

They were then asked to provide a free response to the question: 'What do you think is being measured by the electrodes?'. Finally, they were asked to report whether they thought the experimenter expects this measurement to be greater in the synchronous or asynchronous condition.

### 3.1.4. Measures

In accordance with pre-registered criteria, only participants who gave free responses related to changes in an emotional state (fear or nervous reaction) or a physiological state (e.g. sweat, pulse) relating to fear or nervous reaction were included in SCR expectancy analysis. These criteria were chosen because SCR is considered to indirectly reflect changes in emotional states and is sensitive to physiological change. The number of correct and incorrect answers was independently assessed by two experimenters. Analyses are presented for each of the two raters (PL and ZD) and for the full sample.

### 3.1.5. Pre-registered analyses

Analyses are pre-registered at https://osf.io/tm6aw. Analyses for proprioceptive drift and subjective report expectancies were as in Study 1. For SCR, as pre-registered, we estimated the proportion of correct and incorrect answers to the question regarding what the experimenter expects for SCR with a 95% credibility interval calculated using the Keisan Online Calculator at https://keisan.casio.com/exec/system/1180573226. A Bayes factor modelled with a Bernouilli likelihood with prior beta(1,1) and posterior distribution beta(1 +c,1 + i), with (c) number of correct answers and (i) number of incorrect answers (the prior is a uniform using a Beta distribution, the standard distribution for estimation in a binomial situation, e.g. [32]). Participants also reported whether they expect this measure to be greater following synchronous or asynchronous stroking. A Bayes factor was calculated for the hypothesis that synchronous condition will be chosen with greater frequency than asynchronous condition based on the posterior beta(1 + s, 1 + a), with (s) the number saying synchronous, and (a) the number saying asynchronous. The Bayes factor is the area of the posterior above 0.5, divided by the area below 0.5. This first Bayes factor presumes there is an effect and tests which direction it goes in. A second Bayes factor was calculated to test for the predicted effect versus

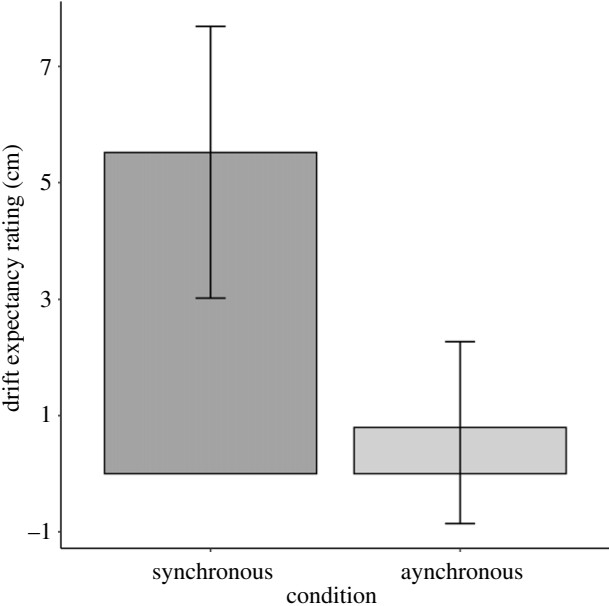

**Figure 4.** Mean proprioceptive drift expectancies for synchronous and asynchronous conditions. Error bars show 95% CIs.

H0. This was calculated using the Rouder binomial calculator at http://pcl.missouri.edu/bf-binomial with H1 modelled as a beta(20,10) because this puts almost all mass above 0.5 (see [33], Appendix 12.2).

### 3.1.6. Exploratory analyses

To investigate whether or not expectancy effects are also seen in people who have not had training in the design of psychological experiments, participants were asked whether or not they had degree-level training in psychology and their responses were used to identify a sub-group of participants without such training for t-test analysis of drift and subjective report expectancies.

To investigate whether a 'control' expectancy statement (which involves an experience of drifting toward the hand) was greater when preceded by judgements of expected perceived hand position, we compared mean C1 synchronous condition expectancies in the combined data from Study 1 and Study 2 ($n = 185$) and data from Lush [11], in which proprioceptive drift expectancies were not measured ($n = 24$).

Bayes factors for exploratory analyses were modelled as for pre-registered analyses. Data, analysis files and analysis output (including boxplots and violin plots) are available at https://osf.io/ct7qe/.

## 3.2. Results

Figure 4 shows proprioceptive drift expectancy ratings for synchronous and asynchronous conditions in the whole sample. Replicating Study 1, proprioceptive drift expectancy was greater for the synchronous condition ($M = 5.5$ cm, s.d. $= 7.8$ cm) than for the asynchronous condition ($M = 0.8$ cm, s.d. $= 5.5$ cm), $t_{44} = 4.61$, $p < 0.001$, 95% CI [2.7, 6.8], $d = 0.69$ 95% CI [0.36, 1.01], $B_{H(0,1)} = 234.43$ $RR_{B>3}$ [0.26, >30].

Figure 5a shows whole sample mean illusion and control statement expectancy ratings for synchronous and asynchronous conditions, and figure 5b shows expectancies for individual statements. As in Study 1 and in Lush [11], 'Illusion' expectancy ratings were greater for the synchronous condition ($M = 1.4$, s.d. $= 1.3$) than for the asynchronous condition ($M = 0.1$, s.d. $= 1.3$), $t_{44} = 5.26$, 95% CI [0.8, 1.7], $d = 0.79$ 95% CI [0.45, 1.12], $B_{H(0,1)} = 2.13 \times 10^5$ $RR_{B>3}$ [0.06, >6]. Synchronous condition 'illusion' expectancy ratings were also greater than synchronous condition control statement expectancy ratings ($M = 0.3$, s.d. $= 1.0$), $t_{44} = 6.11$, $p < 0.001$, 95% CI [0.7, 1.4], $d = 0.91$ 95% CI [0.56, 1.26], $B_{H(0,1)} = 1.88 \times 10^7$ $RR_{B>3}$ [0.04, >6].

Replicating Study 1 results, in the 33 participants who reported not having heard of the procedure previously, proprioceptive drift expectancy was greater for the synchronous condition ($M = 4.8$ cm, s.d. $= 8.5$) than for the asynchronous condition ($M = 0.4$ cm, s.d. $= 5.9$), $t_{32} = 3.78$, $p < 0.001$, 95% CI [2.0, 6.1], $d = 0.66$ 95% CI [0.28, 1.03], $B_{H(0,1)} = 30.76$. $RR_{B>3}$ [0.36, >30].

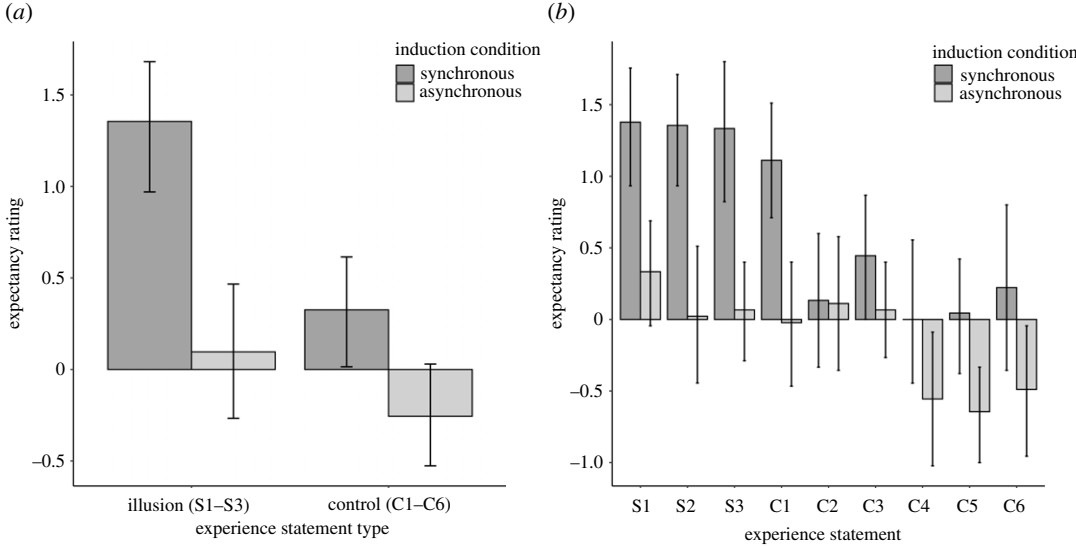

**Figure 5.** (*a*) Subjective report expectancies for the mean illusion (S1–S3) and control (C1–C6) statements for synchronous and asynchronous conditions. Error bars show 95% CIs. (*b*) Subjective report expectancies for individual statements for synchronous and asynchronous conditions. Error bars show 95% CIs.

Illusion expectancy ratings were greater for the synchronous condition ($M = 1.1$, s.d. = 1.4) than for the asynchronous condition ($M = -0.1$, s.d. = 1.2), $t_{32} = 4.54$, $p < 0.001$, 95% CI [0.7, 1.8], $d = 0.79$ 95% CI [0.39, 1.18], $B_{H(0,1)} = 8284.27$ $RR_{B>3}$ [0.07, >30]. Synchronous illusion expectancy ratings were also greater than synchronous control statement expectancy ratings ($M = 0.2$, s.d. = 0.9), $t_{32} = 4.60$, $p < 0.001$ 95% CI [0.5, 1.4], $d = 0.80$ 95% CI [0.40, 1.19]. $B_{H(0,1)} = 9141.29$ $RR_{B>3}$ [0.06, >30].

For the first rater, 25 participants (55.6%) were classified as giving correct answers for the skin conductance question. Of these, 23 thought the experimenter would expect a greater measure in the synchronous condition, compared to two in the asynchronous condition. The proportion reporting synchronous was therefore 0.92 95% CI [0.75, 0.97]. The Bayes factor testing for the direction of the effect assuming there was some effect was $B_{U[0.5,1]\ vs\ U[0,\ 0.5)} = 190838.69$ in favour of more people thinking the effect would be greater in the synchronous than asynchronous direction. The Bayes factor testing this directional effect against H0 was $B_{B(20,10)} = 1632.85$ in favour of the effect.

The second rater interpreted the criteria more liberally and classified 37 participants' (82%) free responses as correct. Of these, 34 reported synchronous and 3 asynchronous as the condition they expected the measure to be greater for. The proportion reporting synchronous was therefore 0.92 95% CI [0.79, 0.97]. The Bayes factor testing for the direction of the effect assuming there was some effect was $B_{U[0.5,1]\ vs\ U(0,0.5)} = 2.99 \times 10^7$ in favour of more people thinking the effect would be greater in the synchronous than asynchronous direction. The Bayes factor testing this directional effect against H0 was $B_{B(20,10)} = 1.03 \times 10^5$ in favour of the effect.

In the whole sample, 41 participants expected a greater SCR effect in the synchronous condition and 4 in the synchronous condition. The proportion reporting synchronous was therefore 0.91, 95% CI [0.79, 0.96]. The Bayes factor testing for the direction of the effect assuming there was some effect was $B_{U[0.5,1]\ vs\ U(0,0.5)} = 3.92 \times 10^8$ in favour of more people thinking the effect would be greater in the synchronous than asynchronous direction. The Bayes factor testing this directional effect against H0 was $B_{B(20,10)} = 1.10 \times 10^6$ in favour of the effect.

### 3.2.1. Exploratory results

In the 31 participants who reported both never having studied psychology at degree level and also never having heard of the presented procedure, synchronous condition drift expectancy ratings ($M = 4.5$ cm, s.d. = 8.6) were greater than asynchronous drift expectancy ratings ($M = 0.3$, s.d. = 6.1) $t_{30} = 3.46$ 95% CI [1.7, 6.7] $p = 0.002$ d = 0.62 95% CI [0.23, 1.00]. $B_{H(0,1)} = 18.67$ RR [0.40, >30]

Illusion expectancy ratings for synchrony ($M = 1.1$, s.d. = 1.4) were greater than for asynchrony ($M = -0.1$, s.d. = 1.2), $t_{30} = 4.33$, $p < 0.001$, 95% CI [0.6, 1.8], $d = 0.78$ 95% 95% CI [0.37, 1.18] $B_{H(0,1)} = 884.90$ $RR_{B>3}$ [0.09, >6]. Synchronous condition illusion expectancy ratings were also greater than

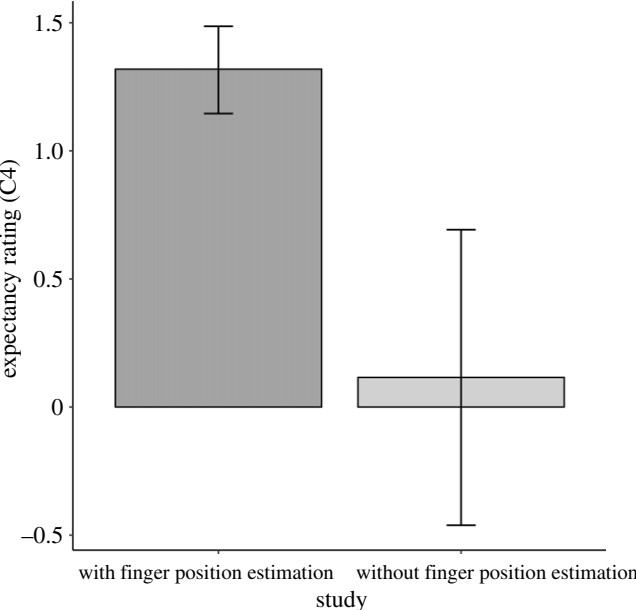

**Figure 6.** Subjective report expectancies for statement C1 'It felt as if my (real) hand were drifting toward the rubber hand' in data from Lush [11] in which proprioceptive drift expectancies were not measured and in combined data from Studies 1 and 2, in which proprioceptive drift expectancies were measured prior to subjective report expectancies. Error bars show 95% CIs.

synchronous condition control ratings ($M = 0.1$, s.d. $= 1.0$), $t_{30} = 4.67$, $p < 0.001$, 95% CI [0.6, 1.5], $d = 0.84$ 95% CI 0.42, 1.24], $B_{H(0,1)} = 6.51 \times 10^4$ $RR_{B>3}$ [0.05, >6].

In Study 1 and Study 2, subjective report expectancies were recorded after the proprioceptive drift procedure, which may have driven expectancies for this statement. Figure 6 shows response to C1 in combined data from Studies 1 and 2 alongside data from Lush [11] in which there was no proprioceptive drift expectancy procedure. Synchronous condition C1 'control' statement expectancies were greater when proprioceptive drift expectancies were measured ($M = 1.1$, s.d. $= 1.4$) than when they were not ($M = 0.1$, s.d. $= 1.6$), $t_{29.22} = 3.67$, $p < 0.001$, 95% CI [0.5, 1.9], $d = 0.84$ 95% 95% CI [0.37, 1.30], $B_{H(0,1)} = RR_{B>3} = 243.91[0.11, >6]$.

# 4. General discussion

In two pre-registered studies, we found that participants reported greater expectancies for synchronous than asynchronous conditions for both indirect measures and subjective report of RH effects. These measures are generally claimed to demonstrate the surprising malleability of the sense of ownership of our own bodies and they support an extensive literature on embodiment [8,34], but participant responses may be attributable to demand characteristics (see also [9,11,17]).

Hypothesis awareness can, of course, confound interpretation of measures. For hypothesis awareness to be controlled (by any reasonable argument), expectancies must be matched across illusion and control conditions. Our results show that this is not the case for asynchronous stroking. We note that there have been attempts to replace or supplement asynchronous stroking as a control method. For example, the invisible hand illusion (in which there is no fake hand and empty space is 'stroked') was originally developed as a control, but was then interpreted as an experimental measure when the authors unexpectedly saw strong responses to it [35]. Future control development requires careful consideration of expectancies prior to deciding on their interpretation. Note that, while controlling for expectancies may deal with confounds of bias and compliance, the consideration of phenomenological control requires an additional step—the control of suggestion difficulty (see [11] for discussion of this issue). This is because even when expectancies are matched, differences between conditions may indicate differences in the ability to generate particular experiences. For effects for which hypothesis unawareness is unlikely (such as RH effects), we have previously proposed a two-stage development process for control methods [11]. First, the control and experimental measures can be matched in expectancies using the kind of procedure reported here. Repeated testing and revision of candidate control statements, for example, might produce well-matched expectancies within a few rounds of

measurement (in different groups of participants). The second step is to match the difficulty of generating the experimental and control experiences by phenomenological control. For this, direct imaginative suggestion can be employed. For example, expectancies might be reasonably well matched for RH effects in response to brushing by a brush and by a laser light (as reported by [22]). It may be that it is more difficult to generate an experience of touch felt on a fake hand when one's own hand is not being touched (by anything other than light) and if so this would not be a good control for a referred touch experience. However, it may be that it is as easy to generate an experience of ownership in response to laser stroking as brush stroking. Stroking with a (unimodal) laser light might therefore provide good control for multisensory integration theories of ownership experience in which touch and vision are considered to jointly drive experience. If difficulty and expectancy are indeed matched, any ownership agreement for brush stroking over and above that seen for laser stroking could be attributed to multisensory integration processes. We note that mechanisms unrelated to demand characteristics might also drive laser light ownership effects, but establishing this would require the development of controls adequate to test the specific mechanism proposed in this case, too.

Consistent with the proposal that proprioceptive drift reflects demand characteristics, Riemer *et al.* [8] note the high variability of drift report, with reports varying from 1 cm to over 5 cm across studies. Riemer *et al.* point out that report varies with different procedures (for example asking participants to point to where they think their hand is rather than verbally report). This highlights the need to test expectancies for different measures. Such variation may arise from differing expectancies across experimental situations and different demand characteristics for different reporting methods. Because most participants expected SCR measures to be greater in the synchronous than in the asynchronous condition, reports of greater SCR in synchronous than asynchronous conditions may also reflect hypothesis awareness effects including faking (e.g. changing facial expression; [25]), imagination (e.g. simulating emotions; [36]) or phenomenological control (see [2]; as mentioned previously, SCR effects have been demonstrated for the direct imaginative suggestion of hand ownership experience; [29]). Altogether, claims that RH measures indicate an illusion which arises from multisensory integration rather than interpretative and creative experience [6,9,10] is not currently supported by evidence from any of subjective report, proprioceptive drift, or SCR measures, either individually or in combination.

In addition to our pre-registered analyses of expectancies for proprioceptive drift and SCR, we conducted an exploratory analysis which further illustrates the influence of demand characteristics in RH effect expectancies. Specifically, expectancies for a control statement describing an experience of the participant's hand drifting towards the RH response were greater following measurement of expectancies regarding perceived hand location (as in the present experiments), than when there was no mention of hand location measurement (as was the case in [11]). Botvinick & Cohen [7] did not measure drift before subjective report and also do not report strong agreement for this control statement. It is plausible that, if in the original study drift had been measured prior to subjective report, this standard control statement would instead be considered a measure of the illusion.

While the present results are confined to the RH paradigm, there are many extensions of this paradigm in which attempts to control demand characteristics also make use of an asynchronous condition. For example, in full body illusions (e.g. [37]) or 'enfacement' (experiences of ownership of and touch referred to another's face e.g. [38]). It is not safe to assume that comparison with asynchronous control conditions is not confounded in these situations either. Measures of the Pinocchio illusion, in which an experience of an elongating nose occurs following stimulation of the bicep while the finger is touching the nose may be attributable to demand characteristics and trait differences in susceptibility to 'perceptual anomalies' [39]. Measures of experimentally induced body 'illusions', beyond the RH effect, therefore require the development of valid controls for demand characteristics and consideration of the influence of trait differences in phenomenological control which are not directly related to embodiment.

There are other indirect measures of RH effects which we have not addressed here. For example, the cross-modal congruency effect in which the participants are asked to report which of two fingers on their own hand has received tactile stimulation while a light is presented on the RH so that its location is congruent or incongruent with the stimulated finger [40]. The difference in reaction time between congruent and incongruent trials is greater for synchronous than asynchronous RH induction. However, differing expectancies may play a role here, too. It has been shown that imagining counterfactual scenarios can change conflict control (for example, imagining words are meaningless in a Stroop task substantially reduces Stroop interference; [41]). Therefore, it is possible that RH cross-modal congruency effects may arise from imagining the fake hand is your own hand in the synchronous condition, but not in the asynchronous condition.

Note that other recent methods (e.g. 2AFC; [42]) may also reflect expectancy effects; a participant who expects to experience an effect for synchronous and not asynchronous stroking can be expected to respond affirmatively when asked if they experience ownership when multisensory signals are relatively synchronous and negatively when they are less synchronous (participant hypothesis awareness in the Chancel *et al.* study was likely, given pre-screening for the report of RH effects). Reducing the possible response to two forced choices (i.e. yes and no) offers no protection against hypothesis awareness in RH effects (see also [12,43]).

Individual differences in the tendency to respond to demand characteristics have been underexplored. Research into this question requires consideration of various aspects of demand characteristics [44]. For example, children may be less likely to form accurate hypothesis awareness, but more likely to react to hypothesis awareness or (hypothesis-mistaken beliefs) with a particular effect (e.g. imagination or phenomenological control). This is a relevant consideration for RH effects, which are often studied in children [45]. Children have been shown to respond to imaginative suggestion to a greater degree than adults (e.g. [46]) and so in the presence of hypothesis awareness, children might be expected to exhibit stronger phenomenological control effects than adults. It seems reasonable also to assume imaginative response to be relatively high in children. As for faking, Brenner [47] argues that, because children as young as four can show awareness of faking emotion, faking cannot be ruled out in mood induction studies of young children. If children are aware that a given response could be faked, it seems plausible that some may fake that response. As a side note, the use of 'implicit' to describe indirect measures can lead to misinterpretation beyond unsafe assumptions that indirect measures are resistant to demand characteristics. For example, in sense of agency research, intentional binding (an indirect measure of sense of agency) is often misinterpreted as a measure of implicit sense of agency. It is a small step from describing something as an implicit method to describing it as a measure of something implicit, but the meanings are very different. We recommend the term 'indirect' rather than 'implicit' to reduce the risk of such errors. The present study illustrates that indirect measures are not necessarily any less vulnerable to demand characteristics than direct subjective report. In summary, the comparison of synchronous and asynchronous conditions in RH effects is confounded by hypothesis awareness. As with subjective reports of experience of referred touch and ownership of a fake hand [11], indirect measures of response to synchronous brush stroking of fake and real hands may reflect compliance, bias and phenomenological control effects rather than (or in addition to) multisensory integration mechanisms.

Ethics. Ethical approval was received from the University of Sussex Sciences & Technology Cross-Schools Research Ethics Committee ('Participant expectancies in the rubber hand illusion': ER/PL207/21; 'amendment 1': ER/PL207/24; 'amendment 2': ER/PL207/26) and informed consent was obtained.

Data accessibility. Data, JASP analysis files and output, and materials are available at: https://osf.io/ct7qe/

Authors' contributions. P.L. and Z.D. designed the study. P.L. collected data, performed analyses and drafted the manuscript. All authors provided critical revisions.

Competing interests. We declare we have no competing interests.

Funding. P.L. and A.K.S. are grateful to the Dr. Mortimer and Theresa Sackler Foundation and to the Canadian Institute for Advanced Research (CIFAR) Program on Brain, Mind and Consciousness.

Acknowledgements. Figure illustration by Julie McDermott.

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
