## [Peer Review File · Royal Society Open Science]

Review History

RSOS-210911.R0 (Original submission)

Review form: Reviewer 1 (Xaver Fuchs)

Is the manuscript scientifically sound in its present form?

Yes

Are the interpretations and conclusions justified by the results?

No

Is the language acceptable?

Yes

Do you have any ethical concerns with this paper?

No

Have you any concerns about statistical analyses in this paper?

No

Recommendation?

Major revision is needed (please make suggestions in comments)

Comments to the Author(s)

The authors present an interesting paper on how implicit measures used in the rubber hand illusion (RHI) literature, namely proprioceptive drift, and skin conductance responses to threat (SCR), might be confounded by demand characteristics. The article uses a similar approach as the study by Lush (2020). That is, participants were asked about what they believe the experimenters intend to induce in the participants but without them actually being tested in the paradigm itself. The study assessed such responses from participants both with respect to the synchronous RHI and the often-used asynchronous control condition and assessed them for questionnaires, proprioceptive drift, and SCR.

As the participants were able to guess that proprioceptive judgments are biased towards the position of the artificial hand and SCR responses higher in the synchronous condition, the authors conclude that even these „implicit“ measures might be confounded by demand characteristics in RHI studies and that control procedures are invalid.

The studies were pre-registered, which is great, and are generally well-performed and the data analyzed appropriately.

I have one major concern about the conclusions of the paper and the methodological approach of the studies.

The studies show that participants, when instructed to think about the experiments and conditions, can guess what the RHI conditions and „implicit“ measures aim to produce in the participants. And their guesses are (at least for questionnaires and proprioceptive drift) very similar to how real data look like. That is interesting and it is potentially problematic for the interpretation of RHI studies and measures because it shows that results might be due to demand characteristics. However, whether this is really the case or not, the authors do not actually show with their studies because the participants did not actually perform the RHI paradigms and measures. While it can be a big problem if participants understand the intention of an experiment, it does not necessarily have to be problematic.

I will give three examples: (1) For example in a well-conducted psychophysical experiment it might be extremely hard for a participant to fake even if s/he understands what the task is about. (2) Participants could be resistant to the demand characteristics and answer very faithfully although being aware of the procedures aims. (3) Admittedly not likely, yet possible, (some) participants might even behave paradoxically to the demand characteristics and behave just the opposite as expected, out of reactance. I admit, the latter ideas are not extremely likely to be true, but they need to be ruled out.

So, I believe, in order to really support the claims of the studies, the authors need to show that the demand characteristics really contaminate the RHI measures. I suggest the authors run two additional experiments where participants first perform the ratings (like done in the presented article) and then (without feedback about whether that is correct) they assess proprioceptive drift and SCR from participants afterward. If there is a high correlation between the ratings and the actual behavior, I would be convinced that the demand characteristics are a big problem and that the authors' conclusions hold. Another approach could be to manipulate participants' expectations about the outcomes of certain measures. In one way or another, I think the inclusion of such data would make the article much stronger.

I have some additional minor points about the manuscript:

P3, L14 (abstract): why is „illusion“ in inverted commas? Another instance of that is in the introduction. If this is to say that it is not actually an illusion, this should be more explicitly expressed

P3, L51: multisensory is sometimes written with a hyphen, sometimes without

P4, L25ff: I think it should be added to the description of the RHI that the real hand is hidden

P4, L44: The authors use the word „substantially“. I think this should be supported by a statistic or further explanations as to why it is substantial

P5, L6ff: I think a paragraph is lacking here that briefly describes what RHI studies generally report before introducing the potential confounds. Otherwise, readers who have not read the previous papers might not be able to follow

P5, L68: I think a solitary number of google results is not a good index of whether the procedure is popular. No doubt at all from my side that it is, I am just skeptical about the piece of evidence here

P6, L30ff: It is also often argued that the tactile input is not that important. The multisensory mechanism at play might be visuo-proprioceptive integration that does not depend on touch and can be observed in many situations using different stimuli or even just vision with no touch at all. Integration can take place here as well so I am not convinced the quoted studies are any proof that drift is a suggestion effect

P6, L50ff: if the authors say there is a „substantial body of research“, they might consider quoting more than just 1-2 studies for this point

P10: L11ff: pls refer to the study from which this procedure has been adapted

P12, L29: Is there a specific reason for which the authors choose the study by Botan et al. (2018) for the estimation of the effect?

P14, Fig 2: It would be more informative to see raw data or distributions, such as violin plots or similar. Some readers might wonder whether participants actually expected a drift and localization at an intermediate position or whether some participants decided for the real hand and others for the rubber hand (leading to a mean value of around 5). In order for the data to look like real proprioceptive drift data, the data should be similar in terms of their distribution, not just mean and sd

P19, L37: It is not clear right away why participants were selected that responses relating to change in arousal etc. This should be made a bit clearer.

P19, L42: Pls give more details about the raters. The manuscript states the raters were „experimenters“, but the experimenters were probably aware of the hypotheses and might be (involuntarily) biased. Or were these blinded raters?

P20, L10ff: it is difficult to follow the rationale of the Bayesian approach here. Some more details might be helpful. Are the authors creating a distribution to serve sort of the null hypothesis that the participants are guessing instead of knowing? Is there a reference for this approach?

P21, Fig 4: as before, I propose to show the data / their distribution. The drift values are very high compared to real data. This should be discussed in the discussion section. It would be interesting if these values were systematically higher than data from the behavioral task.

P24, L9: Pls also express the 25 participants as a percentage

P27, L42: I think that the authors neglect the role of proprioception in the RHI in their argumentation here. Even if there is no tactile input as in the variant introduced by Durgin et al. there is a multisensory conflict between vision and proprioception. Some studies show that merely showing a rubber hand without any stimulation also leads to RHI effects (although weaker). This can, however, still be seen as a visual-proprioceptive conflict (this point is related to the one above concerning the introduction).

P27, L47: I think this interesting discussion about better control procedures could be deepened a bit. How else could demand characteristics be better controlled for?

P28, L11ff: As discussed in Riemer et al. the variation also comes from different methods used (e.g. visual judgments and motor behavior). The authors could consider adding some thought about how different measures of proprioceptive drift can be influenced by demand characteristics. They have chosen a particular one with visual judgments. And the situation in their scenario is slightly artificial. In actual RHI studies, participants do usually have no idea about where their hand and where the RHI is (at least that is the attempt, in fact, they might have a rough idea). However, especially during motor judgments with closed eyes, participants find it quite hard to estimate their hand position and my feeling is that it would actually not be that easy to generate a drift here due to demand characteristics either. So I could imagine that the degree of contamination by demand characteristics might also depend on the exact procedure.

P28, L20ff: I find the discussion of the SCR effects a bit short. I think it would be helpful to repeat the main findings and discuss how this is problematic to RHI studies using SCR

P29, L53ff: I think this point should be discussed in more detail. How can a properly designed 2AFC performance task (such as the cross-model congruency task) be influenced by demand characteristics?

Respectfully reviewed and signed,
Xaver Fuchs

Review form: Reviewer 2

Is the manuscript scientifically sound in its present form?

Yes

Are the interpretations and conclusions justified by the results?

Yes

Is the language acceptable?

Yes

Do you have any ethical concerns with this paper?

No

Have you any concerns about statistical analyses in this paper?

No

Recommendation?

Accept with minor revision (please list in comments)

Comments to the Author(s)

Demand characteristics confound asynchronous control conditions in indirect measures of the rubber hand illusion

Lush and colleagues

In the present manuscript authors investigated the role of demand characteristics in the measures used in a Rubber Hand Illusion (RHI) paradigm, specifically focusing on the comparison between synchronous and asynchronous stimulation conditions. They start from the hypothesis that the differences that typically emerge between synchronous and asynchronous conditions when measured using the proprioceptive drift and skin conductance response (SCR) may reflect response to demand characteristics. In two pre-registered study they tested 185 individuals regarding their expectancies for synchronous and asynchronous conditions measured by the proprioceptive drift and SCR. Results show greater expectancy measures for synchronous compared to asynchronous stimulation conditions in the two studies. Authors interpreted these results as evidence that indirect measures of the RHI may reflect compliance, bias and phenomenological control in response to demand characteristics. Moreover, they suggest that the asynchronous control condition typically used in the RHI paradigm is not a valid control for demand characteristics.

This is a very interesting study that unveil some important concerns regarding the current interpretations that are commonly given to measurements performed in the RHI paradigm.

Specifically, I believe that their consideration on the fact that the so-called objective measures (e.g., proprioceptive drift) cannot be considered as such is a point supported by their data. They also suggested to use the term “indirect” rather than “implicit” to define such measures, which I believe it is a sensible thing to do. I do have some minor comments that authors may want to consider in a revised version of their work that I report below.

Page 5. Authors stated: “In a previous investigation of demand characteristics in RH effects, Tame` et al. (2018) reported a greater magnitude of drift (and subjective report) when participants were asked to report where their hand “really is” rather than where it “feels like” it is, and argued that proprioceptive drift may be attributable to demand characteristics.” I believe that here there is a typo, this was the other way around, greater drift for “feeling” compared to “belief” condition.

I was not able to find the SCR results of the participants. Did authors record this data? It would be interesting to compare the expectancy regarding the SCR data with the actual SCR recorded data.

Authors briefly mentioned what it could be an alternative way of testing to control for the demand characteristics in the RHI paradigm. I was wondering whether they could elaborate more on this aspect, which, I believe is particularly important given the results of their study.

I was wondering to what extent demand characteristics in the RHI paradigm should apply to experiments that include the testing of children. I am not sure if there are studies investigating demand characteristics in children vs adults. Intuitively I would say that children should be less affected by demand characteristics. In an RHI paradigm, some studies have shown smaller drift at a young age (e.g., Nava et al., 2017, Psych Science) though this pattern may vary (e.g., Cowie et al., 2013, Psych Science) compared to adults. Due to the multisensory integration mechanisms not fully developed in young children it may be difficult to distinguish between this factor (multisensory integration) and the demand characteristics. However, I was wondering whether authors can comment on this aspect and whether testing different populations may be useful to provide a better estimate of the factors contributing to the effects measured.

Decision letter (RSOS-210911.R0)

Dear Dr Lush

The Editors assigned to your paper RSOS-210911 "Demand characteristics confound asynchronous control conditions in indirect measures of the rubber hand illusion." have now received comments from reviewers and would like you to revise the paper in accordance with the reviewer comments and any comments from the Editors. Please note this decision does not guarantee eventual acceptance.

Please submit your revised manuscript and required files (see below) no later than 21 days from today's (ie 03-Aug-2021) date. Note: the ScholarOne system will 'lock' if submission of the revision is attempted 21 or more days after the deadline. If you do not think you will be able to meet this deadline please contact the editorial office immediately.

on behalf of Dr Rochelle Ackerley (Associate Editor) and Essi Viding (Subject Editor)
openscience@royalsociety.org

Associate Editor Comments to Author (Dr Rochelle Ackerley):

The reviewers both agree that your manuscript is interesting and has been well-performed. They have provided feedback and comments that need answering, but these would improve the paper further.

Reviewer comments to Author:

Reviewer: 1

Comments to the Author(s)

The authors present an interesting paper on how implicit measures used in the rubber hand illusion (RHI) literature, namely proprioceptive drift, and skin conductance responses to threat (SCR), might be confounded by demand characteristics. The article uses a similar approach as the study by Lush (2020). That is, participants were asked about what they believe the experimenters intend to induce in the participants but without them actually being tested in the paradigm itself. The study assessed such responses from participants both with respect to the synchronous RHI and the often-used asynchronous control condition and assessed them for questionnaires, proprioceptive drift, and SCR.

As the participants were able to guess that proprioceptive judgments are biased towards the position of the artificial hand and SCR responses higher in the synchronous condition, the authors conclude that even these „implicit“ measures might be confounded by demand characteristics in RHI studies and that control procedures are invalid.

The studies were pre-registered, which is great, and are generally well-performed and the data analyzed appropriately.

I have one major concern about the conclusions of the paper and the methodological approach of the studies.

The studies show that participants, when instructed to think about the experiments and conditions, can guess what the RHI conditions and „implicit“ measures aim to produce in the participants. And their guesses are (at least for questionnaires and proprioceptive drift) very similar to how real data look like. That is interesting and it is potentially problematic for the interpretation of RHI studies and measures because it shows that results might be due to demand characteristics. However, whether this is really the case or not, the authors do not actually show with their studies because the participants did not actually perform the RHI paradigms and measures. While it can be a big problem if participants understand the intention of an experiment, it does not necessarily have to be problematic.

I will give three examples: (1) For example in a well-conducted psychophysical experiment it might be extremely hard for a participant to fake even if s/he understands what the task is about. (2) Participants could be resistant to the demand characteristics and answer very faithfully although being aware of the procedures aims. (3) Admittedly not likely, yet possible, (some) participants might even behave paradoxically to the demand characteristics and behave just the opposite as expected, out of reactance. I admit, the latter ideas are not extremely likely to be true, but they need to be ruled out.

So, I believe, in order to really support the claims of the studies, the authors need to show that the demand characteristics really contaminate the RHI measures. I suggest the authors run two additional experiments where participants first perform the ratings (like done in the presented article) and then (without feedback about whether that is correct) they assess proprioceptive drift and SCR from participants afterward. If there is a high correlation between the ratings and the actual behavior, I would be convinced that the demand characteristics are a big problem and that the authors' conclusions hold. Another approach could be to manipulate participants' expectations about the outcomes of certain measures. In one way or another, I think the inclusion of such data would make the article much stronger.

I have some additional minor points about the manuscript:

P3, L14 (abstract): why is „illusion“ in inverted commas? Another instance of that is in the introduction. If this is to say that it is not actually an illusion, this should be more explicitly expressed

P3, L51: multisensory is sometimes written with a hyphen, sometimes without

P4, L25ff: I think it should be added to the description of the RHI that the real hand is hidden

P4, L44: The authors use the word „substantially“. I think this should be supported by a statistic or further explanations as to why it is substantial

P5, L6ff: I think a paragraph is lacking here that briefly describes what RHI studies generally report before introducing the potential confounds. Otherwise, readers who have not read the previous papers might not be able to follow

P5, L68: I think a solitary number of google results is not a good index of whether the procedure is popular. No doubt at all from my side that it is, I am just skeptical about the piece of evidence here

P6, L30ff: It is also often argued that the tactile input is not that important. The multisensory mechanism at play might be visuo-proprioceptive integration that does not depend on touch and can be observed in many situations using different stimuli or even just vision with no touch at all. Integration can take place here as well so I am not convinced the quoted studies are any proof that drift is a suggestion effect

P6, L50ff: if the authors say there is a „substantial body of research“, they might consider quoting more than just 1-2 studies for this point

P10: L11ff: pls refer to the study from which this procedure has been adapted

P12, L29: Is there a specific reason for which the authors choose the study by Botan et al. (2018) for the estimation of the effect?

P14, Fig 2: It would be more informative to see raw data or distributions, such as violin plots or similar. Some readers might wonder whether participants actually expected a drift and localization at an intermediate position or whether some participants decided for the real hand and others for the rubber hand (leading to a mean value of around 5). In order for the data to look like real proprioceptive drift data, the data should be similar in terms of their distribution, not just mean and sd

P19, L37: It is not clear right away why participants were selected that responses relating to change in arousal etc. This should be made a bit clearer.

P19, L42: Pls give more details about the raters. The manuscript states the raters were „experimenters“, but the experimenters were probably aware of the hypotheses and might be (involuntarily) biased. Or were these blinded raters?

P20, L10ff: it is difficult to follow the rationale of the Bayesian approach here. Some more details might be helpful. Are the authors creating a distribution to serve sort of the null hypothesis that the participants are guessing instead of knowing? Is there a reference for this approach?

P21, Fig 4: as before, I propose to show the data / their distribution. The drift values are very high compared to real data. This should be discussed in the discussion section. It would be interesting if these values were systematically higher than data from the behavioral task.

P24, L9: Pls also express the 25 participants as a percentage

P27, L42: I think that the authors neglect the role of proprioception in the RHI in their argumentation here. Even if there is no tactile input as in the variant introduced by Durgin et al. there is a multisensory conflict between vision and proprioception. Some studies show that merely showing a rubber hand without any stimulation also leads to RHI effects (although weaker). This can, however, still be seen as a visual-proprioceptive conflict (this point is related to the one above concerning the introduction).

P27, L47: I think this interesting discussion about better control procedures could be deepened a bit. How else could demand characteristics be better controlled for?

P28, L11ff: As discussed in Riemer et al. the variation also comes from different methods used (e.g. visual judgments and motor behavior). The authors could consider adding some thought about how different measures of proprioceptive drift can be influenced by demand characteristics. They have chosen a particular one with visual judgments. And the situation in their scenario is slightly artificial. In actual RHI studies, participants do usually have no idea about where their hand and where the RHI is (at least that is the attempt, in fact, they might have a rough idea). However, especially during motor judgments with closed eyes, participants find it quite hard to estimate their hand position and my feeling is that it would actually not be that easy to generate a drift here due to demand characteristics either. So I could imagine that the degree of contamination by demand characteristics might also depend on the exact procedure.

P28, L20ff: I find the discussion of the SCR effects a bit short. I think it would be helpful to repeat the main findings and discuss how this is problematic to RHI studies using SCR

P29, L53ff: I think this point should be discussed in more detail. How can a properly designed 2AFC performance task (such as the cross-model congruency task) be influenced by demand characteristics?

Respectfully reviewed and signed,
Xaver Fuchs

Reviewer: 2

Comments to the Author(s)

Demand characteristics confound asynchronous control conditions in indirect measures of the rubber hand illusion

Lush and colleagues

In the present manuscript authors investigated the role of demand characteristics in the measures used in a Rubber Hand Illusion (RHI) paradigm, specifically focusing on the comparison between synchronous and asynchronous stimulation conditions. They start from the hypothesis that the differences that typically emerge between synchronous and asynchronous conditions when measured using the proprioceptive drift and skin conductance response (SCR) may reflect response to demand characteristics. In two pre-registered study they tested 185 individuals regarding their expectancies for synchronous and asynchronous conditions measured by the proprioceptive drift and SCR. Results show greater expectancy measures for synchronous compared to asynchronous stimulation conditions in the two studies. Authors interpreted these results as evidence that indirect measures of the RHI may reflect compliance, bias and phenomenological control in response to demand characteristics. Moreover, they suggest that the asynchronous control condition typically used in the RHI paradigm is not a valid control for demand characteristics.

This is a very interesting study that unveil some important concerns regarding the current interpretations that are commonly given to measurements performed in the RHI paradigm. Specifically, I believe that their consideration on the fact that the so-called objective measures (e.g., proprioceptive drift) cannot be considered as such is a point supported by their data. They also suggested to use the term “indirect” rather than “implicit” to define such measures, which I believe it is a sensible thing to do. I do have some minor comments that authors may want to consider in a revised version of their work that I report below.

Page 5. Authors stated: “In a previous investigation of demand characteristics in RH effects, Tamè et al. (2018) reported a greater magnitude of drift (and subjective report) when participants were asked to report where their hand “really is” rather than where it “feels like” it is, and argued that proprioceptive drift may be attributable to demand characteristics.” I believe that here there is a typo, this was the other way around, greater drift for “feeling” compared to “belief” condition.

I was not able to find the SCR results of the participants. Did authors record this data? It would be interesting to compare the expectancy regarding the SCR data with the actual SCR recorded data.

Authors briefly mentioned what it could be an alternative way of testing to control for the demand characteristics in the RHI paradigm. I was wondering whether they could elaborate more on this aspect, which, I believe is particularly important given the results of their study.

I was wondering to what extent demand characteristics in the RHI paradigm should apply to experiments that include the testing of children. I am not sure if there are studies investigating demand characteristics in children vs adults. Intuitively I would say that children should be less affected by demand characteristics. In an RHI paradigm, some studies have shown smaller drift at a young age (e.g., Nava et al., 2017, Psych Science) though this pattern may vary (e.g., Cowie et al., 2013, Psych Science) compared to adults. Due to the multisensory integration mechanisms not fully developed in young children it may be difficult to distinguish between this factor (multisensory integration) and the demand characteristics. However, I was wondering whether authors can comment on this aspect and whether testing different populations may be useful to provide a better estimate of the factors contributing to the effects measured.

===PREPARING YOUR MANUSCRIPT===

one version identifying all the changes that have been made (for instance, in coloured highlight, in bold text, or tracked changes);
 a 'clean' version of the new manuscript that incorporates the changes made, but does not highlight them. This version will be used for typesetting if your manuscript is accepted.

===PREPARING YOUR REVISION IN SCHOLARONE===

- Any electronic supplementary material (ESM).
- If you are requesting a discretionary waiver for the article processing charge, the waiver form must be included at this step.
- If you are providing image files for potential cover images, please upload these at this step, and inform the editorial office you have done so. You must hold the copyright to any image provided.
- A copy of your point-by-point response to referees and Editors. This will expedite the preparation of your proof.

- Ensure that your data access statement meets the requirements at <https://royalsociety.org/journals/authors/author-guidelines/#data>. You should ensure that you cite the dataset in your reference list. If you have deposited data etc in the Dryad repository, please include both the 'For publication' link and 'For review' link at this stage.
- If you are requesting an article processing charge waiver, you must select the relevant waiver option (if requesting a discretionary waiver, the form should have been uploaded at Step 3 'File upload' above).
- If you have uploaded ESM files, please ensure you follow the guidance at <https://royalsociety.org/journals/authors/author-guidelines/#supplementary-material> to include a suitable title and informative caption. An example of appropriate titling and captioning may be found at https://figshare.com/articles/Table_S2_from_Is_there_a_trade-off_between_peak_performance_and_performance_breadth_across_temperatures_for_aerobic_scope_in_teleost_fishes_/3843624.

Author's Response to Decision Letter for (RSOS-210911.R0)

See Appendix A.

RSOS-210911.R1 (Revision)

Review form: Reviewer 1 (Xaver Fuchs)

Is the manuscript scientifically sound in its present form?

Yes

Are the interpretations and conclusions justified by the results?

Yes

Is the language acceptable?

Yes

Do you have any ethical concerns with this paper?

No

Have you any concerns about statistical analyses in this paper?

No

Recommendation?

Accept as is

Comments to the Author(s)

The authors have submitted a thorough revision and a detailed response letter that addresses my concerns.

The revised manuscript now contains details that were lacking in the previous version and the claims have been adjusted where necessary.

I have no further comments.

Decision letter (RSOS-210911.R1)

Dear Dr Lush,

It is a pleasure to accept your manuscript entitled "Hypothesis awareness confounds asynchronous control conditions in indirect measures of the rubber hand illusion." in its current form for publication in Royal Society Open Science. The comments of the reviewer(s) who reviewed your manuscript are included at the foot of this letter.

on behalf of Dr Rochelle Ackerley (Associate Editor) and Essi Viding (Subject Editor)
openscience@royalsociety.org

Reviewer comments to Author:

Reviewer: 1

Comments to the Author(s)

The authors have submitted a thorough revision and a detailed response letter that addresses my concerns.

The revised manuscript now contains details that were lacking in the previous version and the claims have been adjusted where necessary.

I have no further comments.

Appendix A

Associate Editor Comments to Author (Dr Rochelle Ackerley):

The reviewers both agree that your manuscript is interesting and has been well-performed. They have provided feedback and comments that need answering, but these would improve the paper further.

We thank the editor for the opportunity to revise our manuscript. We address the points raised by the reviewers below.

Reviewer comments to Author:

Reviewer: 1

Comments to the Author(s)

1. The authors present an interesting paper on how implicit measures used in the rubber hand illusion (RHI) literature, namely proprioceptive drift, and skin conductance responses to threat (SCR), might be confounded by demand characteristics. The article uses a similar approach as the study by Lush (2020). That is, participants were asked about what they believe the experimenters intend to induce in the participants but without them actually being tested in the paradigm itself. The study assessed such responses from participants both with respect to the synchronous RHI and the often-used asynchronous control condition and assessed them for questionnaires, proprioceptive drift, and SCR. As the participants were able to guess that proprioceptive judgments are biased towards the position of the artificial hand and SCR responses higher in the synchronous condition, the authors conclude that even these „implicit“ measures might be confounded by demand characteristics in RHI studies and that control procedures are invalid. The studies were pre-registered, which is great, and are generally well-performed and the data analyzed appropriately.

We thank the reviewer for their generous appraisal of our manuscript.

I have one major concern about the conclusions of the paper and the methodological approach of the studies.

The studies show that participants, when instructed to think about the experiments and conditions, can guess what the RHI conditions and „implicit“ measures aim to produce in the participants. And their guesses are (at least for questionnaires and proprioceptive drift) very similar to how real data look like. That is interesting and it is potentially problematic for the interpretation of RHI studies and measures because it shows that results might be due to demand characteristics. However, whether this is really the case or not, the authors do not actually show with their studies because the participants did not actually perform the RHI paradigms and measures. While it can be a big problem if participants understand the intention of an experiment, it does not necessarily have to be problematic.

We have failed to clarify our claim sufficiently. It is not that our results show that RHI measures are due to demand characteristics and, additionally, that control procedures are invalid. Rather the results of our study demonstrate that control procedures are invalid and, consequently, indirect RH measures may be confounded by demand characteristics. It is not our aim, in this paper, to provide evidence that these measures *are* due to demand characteristics (though we have provided evidence elsewhere; e.g., relationships between proprioceptive drift and trait response to imaginative suggestion and order effects ; Lush et al., 2020; Lush, 2021). Rather we aim to show that the control methods used provide no protection against demand characteristics. As a consequence

of the use of invalid control measures, the existing literature on proprioceptive drift and SCR effects in RH studies is uninterpretable because their experimental designs do not distinguish between hypothesis awareness effects (e.g., faking, imagination, and phenomenological control) and other effects.

We have made revisions to the introduction to clarify this:

P7, L11-23: “As Orne (1969) noted, expectancy studies (or “pre-experimental enquiries’) can never provide evidence that a given effect is attributable to demand characteristics. Rather they test the adequacy of an experimental procedure for controlling the effects of demand characteristics. If an experimental procedure is inadequate for controlling, for example, hypothesis awareness, it follows that hypothesis awareness effects cannot be ruled out for studies which employ that procedure. If participant expectancies for synchronous and asynchronous conditions differ in the direction reported in RH experiments, any difference in these measures may be attributable to demand characteristics and consequently, existing reports of proprioceptive drift and SCR may reflect phenomenological control (or other hypothesis awareness effects including imagination and faking; see Corneille & Lush, 2021) rather than, or in addition to, multisensory processes. This would have major implications for interpretation of existing reports of these effects because we would be unable to disentangle hypothesis awareness effects from other effects in any given case.”

We have also added further clarification to P5, L5-14

“An underlying assumption of an experimental control is that it holds everything constant except the independent variable relating to the mechanism of interest. In RH effect studies, the mechanism of interest is typically multisensory integration and the asynchronous control condition is intended keep all factors constant except for the timing of multisensory stimuli. However, participant expectancies may differ for these conditions. If so, this crucial assumption would be violated and the control procedure would not therefore be valid (because any difference between conditions may be attributable to a difference in expectancies instead of or in addition to differences in the timing of multisensory stimuli). It is therefore crucial to establish whether participants have differing expectancies for indirect measures in synchronous and asynchronous conditions.”

2. I will give three examples: (1) For example in a well-conducted psychophysical experiment it might be extremely hard for a participant to fake even if s/he understands what the task is about.

(2) Participants could be resistant to the demand characteristics and answer very faithfully although being aware of the procedures aims.

(3) Admittedly not likely, yet possible, (some) participants might even behave paradoxically to the demand characteristics and behave just the opposite as expected, out of reactance. I admit, the latter ideas are not extremely likely to be true, but they need to be ruled out.

We agree that a wide range of demand characteristic effects are plausible when demand characteristics are uncontrolled, and we appreciate the reviewer raising the important and often overlooked possibility of reactance. We have expanded the manuscript to include discussion of various possible demand characteristics effects. We emphasise that the aim of this paper is not to demonstrate that effects these measures show are due to demand characteristics, but rather to test

the experimental procedure (and therefore the plausibility that demand characteristics confound existing reports).

Added to P3, L 6-12: “Demand characteristics can lead to hypothesis awareness when participant expectancies match experimenters’ predictions. Hypothesis awareness effects include faking and imagination which can generate false positives, but also reactance, which can lead to false negatives (see Corneille & Lush, 2021 for a recent conceptual model of demand characteristics). A particular concern for rubber hand illusion measures is the possibility that they reflect implicit imaginative suggestion effects”

So, I believe, in order to really support the claims of the studies, the authors need to show that the demand characteristics really contaminate the RHI measures. I suggest the authors run two additional experiments where participants first perform the ratings (like done in the presented article) and then (without feedback about whether that is correct) they assess proprioceptive drift and SCR from participants afterward. If there is a high correlation between the ratings and the actual behavior, I would be convinced that the demand characteristics are a big problem and that the authors' conclusions hold. Another approach could be to manipulate participants' expectations about the outcomes of certain measures. In one way or another, I think the inclusion of such data would make the article much stronger.

We agree that such a demonstration would be valuable, and we are currently working on a registered report related to this issue. However, adequately addressing this question will require many hundreds of participants. We believe the results of the current study are important enough to stand on their own, first because there is already evidence consistent with the proposal that judgements of perceived hand position are sensitive to demand characteristics (Tamè et al., 2018; Lush, 2021; Lush et al., 2020), and second because they demonstrate that methods used to support interpretation of a substantial literature are inadequate for that purpose. Demand characteristics are always a threat to interpretation if uncontrolled. This study demonstrates that common RHI procedures do not control for demand characteristics. It cannot be ruled out that, given that people know what is expected of them, that the existing reports of these measures are confounded by faking, imagination, reactance and so on. As previously stated, we do not claim here that there is no effect beyond demand characteristics, merely that it is reasonable to assume that hypothesis awareness may confound existing reports of these effects.

We have also added discussion of a recent manuscript reporting order effects in proprioceptive drift which are consistent with a demand characteristics interpretation P6, L16-20:

“In an exploration of order effects, participants who underwent the asynchronous control condition before the synchronous condition reported a greater difference between synchronous and asynchronous conditions (Lush, 2021). This result may indicate hypothesis awareness arising from task order”

I have some additional minor points about the manuscript:

P3, L14 (abstract): why is „illusion“ in inverted commas? Another instance of that is in the introduction. If this is to say that it is not actually an illusion, this should be more explicitly expressed

We have added text in brackets to the following lines to clarify our position on this issue P3, L 21-22:

“For example, Dieguez (2018) argued that the rubber hand illusion should not be considered an illusion in the same sense as classic visual or optical illusions, as it is likely to arise from

participant expectancies (we agree, and henceforth employ inverted commas when referring to RH effects as illusions),

P3, L51: multisensory is sometimes written with a hyphen, sometimes without

This has been changed to multisensory throughout.

P4, L25ff: I think it should be added to the description of the RHI that the real hand is hidden

Added to P3, L14-16, “Rubber hand (RH) effects involve experiences of ownership and feelings of mislocated touch when a fake hand (which is in view) and the participant’s own hand (which is hidden from view) are brushed in synchrony.”

P4, L44: The authors use the word „substantially“. I think this should be supported by a statistic or further explanations as to why it is substantial

Details of a linear model of the relationship have been added to P4, L1-4:

“For example, trait response to imaginative suggestion on a 6-point scale predicts subjective report of ownership experience on a 7-point scale by 0.8 units per scale point (Lush, Seth & Dienes, 2021; see also Lush et al., 2020; Roseboom & Lush, 2020).”

P5, L6ff: I think a paragraph is lacking here that briefly describes what RHI studies generally report before introducing the potential confounds. Otherwise, readers who have not read the previous papers might not be able to follow

Thank you for this suggestion. Added to P14, L10-17

“In RH research, subjective report is commonly measured by Likert scale responses to three statements describing referred touch and ownership experience, with agreement recorded on a 7-point scale from -3 to +3 (negative values indicate disagreement). These reports are generally taken after the induction procedure has ceased. A set of statements describing other experiences are often included as control statements. See Table 1 for statements and scale labels. Proprioceptive drift is typically measured by reports of the perceived position of the participant’s (unseen) hand before and after the stroking procedure. See Riemer et al (2019) for a review of RH effect procedures.”

P5, L68: I think a solitary number of google results is not a good index of whether the procedure is popular. No doubt at all from my side that it is, I am just skeptical about the piece of evidence here

‘popular’ has been changed to “commonly referenced”

P6, L30ff: It is also often argued that the tactile input is not that important. The multisensory mechanism at play might be visuo-proprioceptive integration that does not depend on touch and can be observed in many situations using different stimuli or even just vision with no touch at all. Integration can take place here as well so I am not convinced the quoted studies are any proof that drift is a suggestion effect

We agree there are other possible explanations, but these also need to be disentangled from hypothesis awareness effects. The following has been added to P6, L12-16:

“Such cases have been interpreted as evidence for multi-modal integration effects either because imagined tactile experience is interpreted as a sensory modality (Durgin et al 2007) or because drift may reflect the integration of proprioceptive and visual information (Rohde et al, 2011). However, these interpretations also require ruling out the influence of demand characteristics.”

Also P31,L16-18:

“We note that mechanisms unrelated to demand characteristics might also drive laser light ownership effects, but establishing this would require development of controls adequate to test the specific mechanism proposed in this case, too.”

P6, L50ff: if the authors say there is a „substantial body of research“, they might consider quoting more than just 1-2 studies for this point

Added P7,L2:

“for a review see Yates (1980)”

Added to P7,L3-4: “including in response to imaginative suggestion (across more than half a century e.g., Barber & Coules, 1959; Kekecs et al., 2016).”

P10: L11ff: pls refer to the study from which this procedure has been adapted

Added to P10,L7:

“(based on the procedure in Botan et al., 2018 and Lush et al, 2020)”

P12, L29: Is there a specific reason for which the authors choose the study by Botan et al. (2018) for the estimation of the effect?

This is the study on which Bayes factors were modelled in our previous rubber hand study, and the prior was chosen for consistency with prior work.

This has been noted , P13,L13:

“and as in Lush et al., 2020”

P14, Fig 2: It would be more informative to see raw data or distributions, such as violin plots or similar. Some readers might wonder whether participants actually expected a drift and localization at an intermediate position or whether some participants decided for the real hand and others for the rubber hand (leading to a mean value of around 5). In order for the data to look like real proprioceptive drift data, the data should be similar in terms of their distribution, not just mean and sd

We would not predict expectancies to perfectly match across an expectancy study and an experimental procedure. The expectancy study demonstrates that it is not safe to assume that participants cannot guess the experimental hypothesis (the expected direction of measures in each condition) in RH studies. If participants show hypothesis awareness, the measure cannot be considered “implicit” and demand characteristics cannot be ruled out. Note also there must be a mechanism by which expectancies are translated into ratings on the actual task (e.g. phenomenological control), and that means the expectancy distributions need not be the same as

the outcome measure distributions. We have added boxplots with violin plots to the open materials and data at <https://osf.io/ct7qe/> or interested readers, but we do not interpret these distributions.

This has been noted on P23,L4-5:

“Data, analysis files and analysis output (including boxplots and violin plots) are available at <https://osf.io/ct7qe/>.”

P19, L37: It is not clear right away why participants were selected that responses relating to change in arousal etc. This should be made a bit clearer.

Added P21, 16-17: “These criteria were chosen because SCR is considered to indirectly reflect changes in emotional states and is sensitive to physiological change.”

P19, L42: Pls give more details about the raters. The manuscript states the raters were „experimenters“, but the experimenters were probably aware of the hypotheses and might be (involuntarily) biased. Or were these blinded raters?

Added to P16,L19: “(PL and ZD)”.

The experimenters were indeed aware of the hypothesis. The results for the full sample are provided to reassure the reader. As noted in the results, just 4 of 45 participants chose the asynchronous condition, so the conclusion holds across all reasonable interpretations of the preregistered criteria.

P20, L10ff: it is difficult to follow the rationale of the Bayesian approach here. Some more details might be helpful. Are the authors creating a distribution to serve sort of the null hypothesis that the participants are guessing instead of knowing? Is there a reference for this approach?

The estimation procedure uses a uniform prior with a Beta distribution, the standard distribution for a binomial situation (e.g. Kruschke & Liddell, 2018); the final test uses an informed prior, as per Dienes (2015, Appendix 12.2).

Added to P23,L4-5: “(the prior is a uniform using a Beta distribution, the standard distribution for estimation in a binomial situation, e.g. Kruschke & Liddell, 2018).”

Added P23,L14: “(see Dienes, 2015, Appendix 12.2).”

P21, Fig 4: as before, I propose to show the data / their distribution. The drift values are very high compared to real data. This should be discussed in the discussion section. It would be interesting if these values were systematically higher than data from the behavioral task.

The procedure itself differs in many ways from a description of the procedure, not least because an intermediary mechanism, such as phenomenological control applies in the actual procedure, and so people may not produce the full effect they wish to produce. The present design can tell us is that it is not safe to assume that participants are not hypothesis aware in drift studies, regarding the expected direction of effect for synchronous and asynchronous conditions. It cannot tell us more than that.

P24, L9: Pls also express the 25 participants as a percentage

Rater classifications have now been expressed as percentages.

P27, L42: I think that the authors neglect the role of proprioception in the RHI in their argumentation here. Even if there is no tactile input as in the variant introduced by Durgin et al. there is a multisensory conflict between vision and proprioception. Some studies show that merely showing a rubber hand without any stimulation also leads to RHI effects (although weaker). This can, however, still be seen as a visual-proprioceptive conflict (this point is related to the one above concerning the introduction).

It seems to us unlikely that matching proprioception and visual input would be sufficient to drive experiences of ownership of the limbs of others. If this were so, standing in a crowd would be a bewildering experience! In any case, the degree to which such effects are attributable to demand characteristics remains an open question.

P27, L47: I think this interesting discussion about better control procedures could be deepened a bit. How else could demand characteristics be better controlled for?

Thank you for this helpful suggestion, which is well aligned with our overall aim to improve experimental design and methodology in studies of this sort. We have added top P30-P31, L22-L18):

“For effects for which hypothesis awareness is likely (such as rubber hand effects), we have previously proposed a two stage development process for control methods (Lush, 2020). First, the control and experimental measures can be matched in expectancies using the kind of procedure reported here. Repeated testing and revision of candidate control statements, for example, might produce well matched expectancies within a few rounds of measurement (in different groups of participants). The second step is to match the difficulty of generating the experimental and control experiences by phenomenological control. For this, direct imaginative suggestion can be employed. For example, expectancies might be reasonably well matched for RH effects in response to brushing by a brush and by a laser light (as reported by Durgin et al, 2007). It may be that it is more difficult to generate an experience of touch felt on a fake hand when one’s own hand is not being touched (by anything other than light) and if so this would not be a good control for referred touch experience. However, it may be that it is as easy to generate an experience of ownership in response to laser stroking as brush stroking. Stroking with a (unimodal) laser light might therefore provide a good control for multisensory integration theories of ownership experience in which touch and vision are considered to jointly drive experience. If difficulty and expectancy are indeed matched, any ownership agreement for brush stroking over and above that seen for laser stroking could be attributed to multisensory integration processes. We note that mechanisms unrelated to demand characteristics might also drive laser light ownership effects, but establishing this would require the development of controls adequate to test the specific mechanism proposed in this case, too.”

P28, L11ff: As discussed in Riemer et al. the variation also comes from different methods used (e.g. visual judgments and motor behavior). The authors could consider adding some thought about how different measures of proprioceptive drift can be influenced by demand characteristics. They have chosen a particular one with visual judgments.

We thank the reviewer for this suggestion and have added the following to P31,L21-25:

“Riemer et al point out that report varies with different procedures (for example asking participants to point to where they think their hand is rather than verbally report). This

highlights the need to test expectancies for different measures. Such variation may arise from differing expectancies across experimental situations and different demand characteristics for different reporting methods”

And the situation in their scenario is slightly artificial. In actual RHI studies, participants do usually have no idea about where their hand and where the RHI is (at least that is the attempt, in fact, they might have a rough idea).

We disagree that participants usually have no idea about where their hand is when it is hidden from sight. Even in the complete absence of proprioceptive information (for example in a deafferented patient) participants would still have knowledge of where they placed their hand, and the position of their hand is constrained by the position of other parts of their body. Additionally, in everyday life people are rather good at directing hand movements without visual feedback – a common public demonstration of this ability is (for example) to ask people to touch their nose with their eyes closed or to touch their two index fingers together. Most people can do this without any difficulty. That aside, the issue at stake is whether participants are able to work out that the experimenter expects them to experience their hand as closer to the fake hand following synchronous stroking (compared to asynchronous stroking). There will always be differences between reading about experiments and participating in experiments. The question is whether or not it is safe to assume that participants cannot work out the hypothesis when participating in an experiment. We believe the design employed here is sufficient to suggest it is not safe to assume hypothesis unawareness in participants.

However, especially during motor judgments with closed eyes, participants find it quite hard to estimate their hand position and my feeling is that it would actually not be that easy to generate a drift here due to demand characteristics either. So I could imagine that the degree of contamination by demand characteristics might also depend on the exact procedure.

While we agree that the degree of influence of demand characteristics might well depend on the exact procedure, all that is required for demand characteristic effects consistent with the experimental hypothesis is that participants have hypothesis awareness. This study shows that, on average, they do. In any given rubber hand study, we simply do not know what percentage of participants are faking (i.e., telling the experimenter what they want to hear), reactant (i.e., telling the experimenter what they do not want to hear), actively imagining their hand has drifted, or experiencing an involuntary relocation of their hand due to phenomenological control. This information is required before we can interpret the results as evidence of multisensory integration mechanisms influencing body ownership (that is, what percentage of participants, if any, are experiencing a relocation of their hand due to matching sensory information from different modalities). This is the case whether tactile information is important or not. If demand characteristics are not controlled, we cannot rule out demand characteristics as an explanation. That is, we cannot safely assume that participants do not respond to demand characteristics, in measuring anomalous experience, or elsewhere.

P28, L20ff: I find the discussion of the SCR effects a bit short. I think it would be helpful to repeat the main findings and discuss how this is problematic to RHI studies using SCR

Added to P31-32, L25-6:

“Because most participants expected SCR measures to be greater in the synchronous than in the asynchronous condition, reports of greater SCR in synchronous than asynchronous conditions may also reflect hypothesis awareness effects including faking (e.g., changing

facial expression; Levenson et al., 1990), imagination (e.g., simulating emotions; Stern & Lewis, 1968), or phenomenological control (see Corneille & Lush, 2021; as mentioned previously, SCR effects have been demonstrated for direct imaginative suggestion of hand ownership experience; Hägni et al., 2008).”

P29, L53ff: I think this point should be discussed in more detail. How can a properly designed 2AFC performance task (such as the cross-modal congruency task) be influenced by demand characteristics?

Added discussion of 2AFC tasks to P33-34, L21-3:

“Note that other recent methods (e.g., 2AFC; Chancel et al., 2021) may also reflect expectancy effects; a participant who expects to experience an effect for synchronous and not asynchronous stroking can be expected to respond affirmatively when asked if they experience ownership when multisensory signals are relatively synchronous and negatively when they are less synchronous (participant hypothesis awareness in the Chancel et al study was likely, given pre-screening for report of RH effects). Reducing the possible response to two forced choices (i.e., yes and no) offers no protection against hypothesis awareness in RH effects (see also Lush et al., 2021; Seth et al., 2021).”

Revised discussion of cross-modal congruency to improve clarity, P33,L15-20:

“It has been shown that imagining counterfactual scenarios can change conflict control (for example, imagining words are meaningless in a Stroop task substantially reduces Stroop interference (Palfi et al., 2021). Therefore, it is possible that RH cross-modal congruency effects may arise from imagining the fake hand is your own hand in the synchronous condition, but not in the asynchronous condition.”

Reviewer: 2

Comments to the Author(s)

Demand characteristics confound asynchronous control conditions in indirect measures of the rubber hand illusion

Lush and colleagues

In the present manuscript authors investigated the role of demand characteristics in the measures used in a Rubber Hand Illusion (RHI) paradigm, specifically focusing on the comparison between synchronous and asynchronous stimulation conditions. They start from the hypothesis that the differences that typically emerge between synchronous and asynchronous conditions when measured using the proprioceptive drift and skin conductance response (SCR) may reflect response to demand characteristics. In two pre-registered study they tested 185 individuals regarding their expectancies for synchronous and asynchronous conditions measured by the proprioceptive drift and SCR. Results show greater expectancy measures for synchronous compared to asynchronous stimulation conditions in the two studies.

Authors interpreted these results as evidence that indirect measures of the RHI may reflect compliance, bias and phenomenological control in response to demand characteristics. Moreover, they suggest that the asynchronous control condition typically used in the RHI paradigm is not a valid control for demand characteristics.

We thank the reviewer for their accurate summary. However, the final point is incorrect. It is not that we interpret these results as indicating response to demand characteristics and additionally that the asynchronous control condition is not a valid control. Rather, this study demonstrates that the the use of an asynchronous control condition is confounded by hypothesis awareness. And as a consequence of this, existing studies which employ this control condition may be confounded by demand characteristics. We have revised the text to clarify this claim in response to R1.

This is a very interesting study that unveil some important concerns regarding the current interpretations that are commonly given to measurements performed in the RHI paradigm. Specifically, I believe that their consideration on the fact that the so-called objective measures (e.g., proprioceptive drift) cannot be considered as such is a point supported by their data. They also suggested to use the term “indirect” rather than “implicit” to define such measures, which I believe it is a sensible thing to do.

We thank the reviewer for this positive evaluation.

I do have some minor comments that authors may want to consider in a revised version of their work that I report below.

Page 5. Authors stated: “In a previous investigation of demand characteristics in RH effects, Tamè et al. (2018) reported a greater magnitude of drift (and subjective report) when participants were asked to report where their hand “really is” rather than where it “feels like” it is, and argued that proprioceptive drift may be attributable to demand characteristics.” I believe that here there is a typo, this was the other way around, greater drift for “feeling” compared to “belief” condition.

We thank the reviewer for noting this error – it has now been corrected.

I was not able to find the SCR results of the participants. Did authors record this data? It would be interesting to compare the expectancy regarding the SCR data with the actual SCR recorded data.

No SCR data were recorded, only expectancies.

Authors briefly mentioned what it could be an alternative way of testing to control for the demand characteristics in the RHI paradigm. I was wondering whether they could elaborate more on this aspect, which, I believe is particularly important given the results of their study.

See our response to R1 on this point.

I was wondering to what extent demand characteristics in the RHI paradigm should apply to experiments that include the testing of children. I am not sure if there are studies investigating demand characteristics in children vs adults. Intuitively I would say that children should be less affected by demand characteristics.

Children 5-8 years old have a hypnotisability about the same as undergraduates; hypnotisability increases until 9-12 then slightly decreases down again for adults (Morgan & Hilgard, 1973).

In an RHI paradigm, some studies have shown smaller drift at a young age (e.g., Nava et al., 2017, Psych Science) though this pattern may vary (e.g., Cowie et al., 2013, Psych Science) compared to adults. Due to the multisensory integration mechanisms not fully developed in young children it may be difficult to distinguish between this factor (multisensory integration) and the demand characteristics. However, I was wondering whether authors can comment on this aspect and whether testing different populations may be useful to provide a better estimate of the factors contributing to the effects measured.

Added to P34,L13-19:

“Individual differences in the tendency to respond to demand characteristics has been underexplored. Research into this question requires consideration of various aspects of demand characteristics (Lush & Cornielle, 2021). For example, children may be less likely to form accurate hypothesis awareness, but more likely to react to hypothesis awareness or (hypothesis-mistaken beliefs) with a particular effect (e.g., imagination or phenomenological control). This is a relevant consideration for RH effects, which are often studied in children (Lee, Ma & Kammers, 2021). Children have been shown to respond to imaginative suggestion to a greater degree than adults (e.g., Barber & Calverley, 1963) and so in the presence of hypothesis awareness, children might be expected to exhibit stronger phenomenological control effects than adults. It seems reasonable also to assume imaginative response to be relatively high in children. As for faking, Brenner (2000) argues that, because children as young as four can show awareness of faking emotion, faking cannot be ruled out in mood induction studies of young children. If children are aware that a given response could be faked, it seems plausible that some may fake that response.”

References

- Barber, T. X., & Calverley, D. S. (1963). 'Hypnotic-like' suggestibility in children and adults. *The Journal of Abnormal and Social Psychology*, 66(6), 589–597.
<https://doi.org/10.1037/h0041709>
- Botan, V., Fan, S., Critchley, H., & Ward, J. (2018). Atypical susceptibility to the rubber hand illusion linked to sensory-localised vicarious pain perception. *Consciousness and Cognition*, 60, 62–71. <https://doi.org/10.1016/j.concog.2018.02.010>
- Brenner, E. (2000). Mood Induction in Children: Methodological Issues and Clinical Implications. *Review of General Psychology*, 4(3), 264–283. <https://doi.org/10.1037/1089-2680.4.3.264>
- Chancel, M., Ehrsson, H. H., & Ma, W. J. (2021). *Uncertainty-based inference of a common cause for body ownership*. OSF Preprints. <https://doi.org/10.31219/osf.io/yh2z7>
- Corneille, O., & Lush, P. (2021). *Sixty years after Orne's American Psychologist article: A conceptual analysis of "Demand Characteristics"*. PsyArXiv. <https://doi.org/10.31234/osf.io/jqyvz>
- Dienes, Z. (2015). How Bayesian statistics are needed to determine whether mental states are unconscious. In *Behavioral Methods in Consciousness Research*. Oxford University Press.
<https://doi.org/10.1093/acprof:oso/9780199688890.003.0012>
- Kekecs, Z., Szekely, A., & Varga, K. (2016). Alterations in electrodermal activity and cardiac parasympathetic tone during hypnosis. *Psychophysiology*, 53(2), 268–277.
<https://doi.org/10.1111/psyp.12570>
- Kruschke, J.K., Liddell, T.M. (2018). The Bayesian New Statistics: Hypothesis testing, estimation, meta-analysis, and power analysis from a Bayesian perspective. *Psychonomic Bulletin & Review*, 25, 178–206. <https://doi.org/10.3758/s13423-016-1221-4>
- Lee, L., Ma, W., & Kammers, M. (2021). The rubber hand illusion in children: What are we measuring? *Behavior Research Methods*. <https://doi.org/10.3758/s13428-021-01600-x>

- Levenson, R. W., Ekman, P., & Friesen, W. V. (1990). Voluntary Facial Action Generates Emotion-Specific Autonomic Nervous System Activity. *Psychophysiology*, 27(4), 363–384.
<https://doi.org/10.1111/j.1469-8986.1990.tb02330.x>
- Lush, P. (2020). Demand Characteristics Confound the Rubber Hand Illusion. *Collabra: Psychology*, 6(1), 22. <https://doi.org/10.1525/collabra.325>
- Lush, P. (2021). *Order effects in the rubber hand illusion*. PsyArXiv.
<https://doi.org/10.31234/osf.io/amsrp>
- Lush, P., Botan, V., Scott, R. B., Seth, A. K., Ward, J., & Dienes, Z. (2020). Trait phenomenological control predicts experience of mirror synaesthesia and the rubber hand illusion. *Nature Communications*, 11(1), 4853. <https://doi.org/10.1038/s41467-020-18591-6>
- Lush, P., Dienes, Z., & Seth, A. (2021). *Rubber hand illusion reports remain confounded by demand characteristics and are substantially related to trait phenomenological control*. PsyArXiv.
<https://doi.org/10.31234/osf.io/qh4ag>
- Morgan, A. H., & Hilgard, E. R. (1973). Age differences in susceptibility to hypnosis. *International Journal of Clinical and Experimental Hypnosis*, 21(2), 78–85.
<https://doi.org/10.1080/00207147308409308>
- Orne, M.T. Demand characteristics and the concept of quasi-controls. In R. Rosenthal & R. Rosnow (Eds.), *Artifact in Behavioral Research*. New York: Academic Press, 1969. Pp. 143-179.
- Seth, A., Roseboom, W., Dienes, Z., & Lush, P. (2021). *What's up with the Rubber Hand Illusion?* PsyArXiv. <https://doi.org/10.31234/osf.io/b4qcy>
- Stern, R. M., & Lewis, N. L. (1968). Ability of Actors to Control Their GSRs and Express Emotions. *Psychophysiology*, 4(3), 294–299. <https://doi.org/10.1111/j.1469-8986.1968.tb02770.x>
- Tamè, L., Linkenauger, S. A., & Longo, M. R. (2018). Dissociation of feeling and belief in the rubber hand illusion. *PLOS ONE*, 13(10), e0206367. <https://doi.org/10.1371/journal.pone.0206367>

Yates, A. J. (1980). Voluntary Control of Autonomic Functions. In A. J. Yates (Ed.), *Biofeedback and the Modification of Behavior* (pp. 160–267). Springer US. https://doi.org/10.1007/978-1-4684-3554-2_4